# Microalgae Oil as an Effective Alternative Source of EPA and DHA for Gilthead Seabream (*Sparus aurata*) Aquaculture

**DOI:** 10.3390/ani11040971

**Published:** 2021-03-31

**Authors:** Ester Santigosa, Fabio Brambilla, Luca Milanese

**Affiliations:** 1DSM Nutritional Products, Research Centre for Animal Nutrition & Health, 68128 Village-Neuf, France; 2VRM Srl Naturalleva, 37137 Verona, Italy; fabio_brambilla@naturalleva.it (F.B.); luca_milanese@naturalleva.it (L.M.)

**Keywords:** microalgae, fish oil replacement, sustainability, EPA DHA alternative source

## Abstract

**Simple Summary:**

Aquaculture must rely on sustainable aquafeeds to attain the future demand for farmed and high-quality seafood. Algae-derived ingredients, particularly oils from microalgae, have recently emerged as an alternative to fish oil and vegetable oils extracted from terrestrial plants. Such microalgae oils are sustainable alternatives, rich in essential fatty acids and free of dioxins and polychlorinated biphenyls (PCBs). This study assesses the incorporation of microalgae oils in the diet of gilthead seabream, which is a carnivorous species with typically high demand for fish-based ingredients, particularly fishmeal and fish oil. In particular, its effect on feed quality, seabream growth performance, fillet quality, and safety were compared when fish were fed a diet rich in fish oil versus diets formulated with microalgal oils (3.5 or 0.7% incorporation rate). Importantly, fish performance was maintained when fish oil was replaced by microalgae oil, and fish fillet showed noticeably lower levels of contamination and similar sensory quality. Collectively, the findings of this study showed that aquafeeds formulated with microalgae oils fulfil important nutritional requirements of gilthead seabream and result in a high quality product. Ultimately, these findings offer a more resilient means for sustaining the future growth of gilthead seabream aquaculture.

**Abstract:**

Microalgal oils (AOs) emerged recently as an alternative to fish oil and to nutritionally poorer vegetable oils for fish species. In this trial, two experimental diets containing fish oil (negative control: 2.1%; positive control: 13.8%) and two diets incorporating AO at 3.5 and 0.7% were fed to grow out gilthead seabream (*Sparus aurata*) of 64.5 g initial body weight. After 110 days of experimental feeding, performance (final body weight mean = 147 g) and survival (>99%) were similar across treatments. The highest eicosapentaenoic acid (EPA) + docosahexaenoic acid (DHA) content in positive control (PC) and 3.5 AO feeds (3.11 and 2.18% of diet, respectively) resulted in the highest EPA + DHA deposition in the fillets (18.40 and 12.36 g/100 g fatty acid, respectively), which entirely reflected the dietary fatty acid profile. Feed and fillets from fish fed the AO diets had lower levels of dioxins and polychlorinated biphenyls (PCBs). Moreover, sensory quality of AO fillets scored equally to the PC fish. Collectively, these findings offer a more resilient means for sustaining the future growth of seabream aquaculture, whilst maintaining the nutritional value of the resulting seafood. The data supports the addition of seabream to the list of aquaculture species where microalgal oil can be used as an ingredient to fulfil their challenging nutritional demands.

## 1. Introduction

The global aquaculture industry is the fastest growing food sector with production already surpassing that of wild-caught seafood and beef [1]. The industry is therefore theorised to be able to best meet the challenge of feeding the growing population [2,3,4]. Importantly, however, these production lines must preserve food quality [5] while improving sustainability [4].

Plant-based diets have been used by the aquaculture industry to replace less sustainable fishmeal-based feeds without any detriment to fish health [6,7]. Yet, the industry still relies on the use of fish oil to grow nutritionally rich, high-quality aquaculture products for human consumption [8,9]. To guarantee both the long-term continued sustainable growth of aquaculture and product quality, alternatives to fish oil are desperately needed: aquafeeds formulated with vegetable oils alone lack or contain lower amounts of key fatty acids highly desirable for a healthy animal and human diet, such as eicosapentaenoic acid (EPA) or docosahexaenoic acid (DHA) [10]. In humans, n-3 fatty acids, and EPA and DHA specifically, are linked to a number of developmental and health benefits [11], including increased cardiovascular integrity [12,13] and the alleviation of inflammatory syndromes, such as rheumatoid arthritis, ulcerative colitis, and atopic dermatitis psoriasis [14,15,16]. The replacement of fishmeal and oils with plant-based alternatives containing low levels of these essential n-3 acids halved the EPA + DHA content of salmon fillets between 2004 and 2015 [17] and reduced DHA altering the EPA:DHA ratio in seabream fillets [18]. In turn, this increases the number of fish portions necessary to obtain dietary recommendations (from 1.75 g weekly; EFSA, 2016 to 3.5 g weekly; International Society for the Study of Fatty Acids and Lipids and the Global Organisation for EPA and DHA). For instance, Sprague et al. [19] showed that salmons obtained from UK retailers fulfilled only 26–67% of the 3.5 g EPA + DHA recommended weekly intake, evidencing a very high nutritional variability in fish nutritional value.

Beyond human health, the skin of fish reared on reduced levels of dietary EPA and DHA loses its functionality as a barrier to pathogens [20] and fillets of such fish often feature melanin spots [21], both suggesting compromised immunological competency of cultured fish. Moreover, the EPA:DHA ratio determines microbiome composition and intestinal health, as well as feed and protein utilisation [22,23]. Therefore, dietary availability of both EPA and DHA has implications for fish growth and survival [23,24,25,26,27]. Due to the complementary and individual health benefits of EPA and DHA [22,28], reversing or halting a decline in content of these n-3 fatty acids found in aquaculture feeds and products is crucial.

Marine microalgae are a natural source of fatty acids, including EPA and DHA [29], which can be easily incorporated into fish feed formulations [29] and contribute to the economic sustainability of aquaculture through de-risking the supply of critical nutrients in the aquaculture feed. Substantial financial risk exists for businesses that deliver sub-optimal nutrition for livestock held under their conditions, because nutritional requirements for sea bream [25] as well as other farmed fish species [30] include minimum requirements for essential fatty acids such as EPA, DHA, and arachidonic acid (ARA). Various studies [17,19,29] have demonstrated that omega-3 levels in farmed aquaculture feeds have fallen over the past decade, and more, as aquaculture has continued to grow whilst the availability of fish oil, the traditional sources of essential fatty acids for aquaculture feed have stagnated [7,9,31]. This not only risks the nutritional quality of the fish as a healthy food for people to eat, but it also puts at risk the health and therefore the welfare of fish held under farming conditions at fish farms all over the world. The supplementation with microalgal oils containing both EPA and DHA can mirror the omega-3 content found in diets based on marine ingredients. This has been shown within rainbow trout with no negative effect on digestibility of the feed and growth of the fish [5]. Crucially, fillets from rainbow trout fed a microalgal oil-enriched diet show an EPA and DHA content comparable to those reared on fish oil and, thus, meet human nutritional requirements [5]. Additionally, using microalgal products reduces contamination of fillets with chemical compounds such as dioxin-like polychlorinated biphenyls (PCBs) or mercury, which typically accumulate in feed made from animals at a higher trophic level, such as fish [32,33].

A number of studies were able to replace fish or vegetable oils with microalgal oil in aquafeeds of salmonids [5,34,35], but until now there have not been any studies to document the efficacy and performance of algal oil (AO) as a replacement oil for grow-out gilthead seabream (*Sparus aurata*) diets. Gilthead seabream is the leading species in finfish aquaculture in the Mediterranean Sea but has a reduced ability to biosynthesise long-chain EPA and DHA from C-18 precursors (linoleic and α-linoleic acid), making dietary supplementation essential [22,36]. For optimal growth, *S. aurata* requires at least 1.3% dietary EPA + DHA [25]. Requirements can be increased if farming conditions are challenging (i.e., low temperatures, disease, handling, etc.). In this sense, a more recent publication [37] has shown a clear growth response curve to n-3 LC-PUFA and optimal growth at 3% EPA + DHA dietary inclusion for this species. Similarly, Rodriguez et al. [38] suggested that a larger proportion of DHA was beneficial for gilthead seabream. However, in a later study of black seabream [39], a higher DHA:EPA ratio resulted in higher malondialdehyde (MDA) levels, indicating an increase in fish oxidative stress. These differences in the literature make this an ideal model species to evaluate the benefits of using aquafeeds supplemented with microalgal oils and different DHA:EPA ratios. Here, we demonstrated the results of an experiment designed to compare the fatty acid profile, tissue integrity, and level of contamination of feed and fillets of gilthead seabream supplied, with aquafeeds formulated using either fish or microalgal oil. In addition, we also assessed whether a microalgal oil-based diet changes the zootechnical performance of the fish, as well as the sensory quality of the product.

## 2. Materials and Methods

### 2.1. Diets and Experimental Design

The basal feed was produced at the feed mill of the VRM Srl Naturalleva plant located in Cologna Veneta (Verona, Italy). Formulations were modified only for lipid fraction, with the aim of replacing fish oil with microalgal oil. The rest of the oils in the formulation were adjusted to keep the diets isolipidic and to help balance the omega3:omega6 ratio, which can be adversely impacted if vegetable oils are not blended. The diet with 2.10% of fish oil functioned as a negative control (NC), while a diet including 13.79% of fish oil and, hence, mimicking a commercial diet, was used as positive control (PC). The treatment group consisted of two more diets formulated to include two levels of Veramaris^®^ microalgal oil (3.5 and 0.7% AO) as the sole source of long-chain polyunsaturated fatty acids (diets 3.5AO and 0.7AO, respectively). The microalgal oil is produced from a heterotrophic algal strain of *Schizochytrium* sp., which synthesises the omega-3 fatty acids EPA and DHA naturally (fatty acid profile of the AO detailed in Table A1). The formula of each of the four diets is described in Table 1. Diets were formulated to contain 93% of dry matter and to be isoenergetic (21.4 MJ/Kg), isonitrogenous (43% crude protein), and isolipidic (18% crude lipid). The mash of each diet was extruded to produce 3-mm pellets using a single-screw extruder (X-165, Wenger, Sabetha KS 66534, USA) before vacuum coating (La Meccanica vacuum coater, Italy). Proximate analyses of the fatty acid profiles expected for each diet were performed on three samples by the VRM Srl Naturalleva internal lab (*n* = 3; Table 1; Table A2).

### 2.2. Rearing Conditions and Zootechnical Measurements

The feeding trial was carried out at the Experimental Center of VRM srl Naturalleva^®^ in Civitavecchia (Italy) in compliance with Directive 2010/63/EU on the protection of animals used for scientific purposes. Eyed gilthead seabream (*Sparus aurata*) eggs were hatched in the facilities in tanks supplied with flow-through water up until their first feeding. They were then reared in 1000-L tanks until the desired initial body weight was reached.

For the experiment, a total of 960 seabream (initial body weight: 64.5 g ± 1.0; mean ± standard deviation) were randomly distributed between 12 fiberglass 2000-L tanks (3 replicates per experimental diet treatment, i.e., *n* = 3; 80 fish per tank). During the week, animals were fed the experimental diets twice a day (morning: 9 am; afternoon: 3pm), while an automatic feeder delivered the feeds on weekends and bank holidays. Diets were administered for 110 days. Unconsumed feed was collected from the bottom of the tank one hour after each meal by siphoning, dried at 70 °C, and weighed to compute feed intake.

The experimental tanks are part of a recirculating unit fed with fresh seawater with three complete daily water renewals in each tank. Oxygen (Handheld Oxymeter WTW OXI 340i) and temperature were monitored daily, whereas pH, ammonia, nitrates, and nitrites were checked biweekly (Mérieux Nutriscience, Italy). The average temperature throughout the experiment was 20.0 °C (±2.8 standard deviation) across all tanks fed with different diets. Fish were bulk weighed every two weeks until the end of the experiment in day 110. Feed intake, and initial and final body weight data allowed us to calculate growth, specific growth rate, and the feed conversion ratio as measures of zootechnical performance.

Specific growth rate (SGR, %BW d^−1^) = [100 × (ln (FBW/IBW))] × d^−1^, where FBW and IBW are final and initial body weight, respectively, and d is the number of days between FBW and IBW;Feed conversation ratio (FCR) = feed intake × (FBW − IBW)^−1^;Net apparent retention (%) = (0.61 × FBW × FFA) − (0.61 × IBW × IFA) × (total feed × fatty acid concentration in feed)^−1^ FFA and IFA represent final and initial fatty acid composition and 0.61 corresponds to the edible portion of the fish.

### 2.3. Analytics

#### 2.3.1. Muscle Full Fatty Acid Profile and Retention

To determine the muscle fatty acid profile and net apparent retention, 20 animals were randomly sampled at the beginning of the trial and 4 animals per tank at the end of the experiment. Animals were anesthetised using 0.08 g of tricaine methanesulfonate (MS-222)/L and sacrificed by a blow on the head. Muscle samples (5 cm × 5 cm) were taken from the left fillet, immediately frozen (−20 °C) and maintained at −20 °C until analysed for lipid and fatty acid composition. The extraction and analysis of muscle lipid was completed using the chloroform–methanol method according to [40]. The fatty acid composition of lipids from diet and pooled fish tissues was analysed at Merieux NutriScience Lab (Italy) via a gas chromatograph using a flame ionisation detector. Fatty acids in lipid samples were esterified first with KOH methanol and then with HCL-methanol in a 72-°C water bath. Fatty acid methyl esters were extracted with hexane and then separated via gas chromatography (GC-2025 Shimadzu, Kyoto, Japan) with a fused silica capillary column (SP2560 100 m × 0.25 mm × 0.2 mm, Merck; 007-CW, Hewlett Packard, Palo Alto, CA, USA) and based on Sigma–Aldrich standards (Supelco 37 Component FAME Mix; PUFA No. 1). The column temperature was set to rise from 100 to 240 °C at a rate of 4 °C/min. Both the injector and detector temperatures were set at 250 °C.

#### 2.3.2. Lipid Peroxidation

To measure oxidative stress, malondialdehyde values (MDA) were determined using the distillation method and a 2-thiobarbituric acid according to [41]. The absorbance values were determined utilising a JANWAY 7315 Spectrophotometer (Bibby Scientific Italia) at 538 nm against a blank containing 5 mL of distilled water and 5 mL of thiobarbituric acid solution. Results were expressed in mg MDA/kg.

#### 2.3.3. Contaminants in Feed and Fish Fillet Samples

Feed and fish fillet samples from the PC diet group and both microalgal oil diet groups (*n* = 12) were analysed by Eurofins (France). Arsenic and mercury were measured with iron-mobility spectrometry-mass spectrometry. Dioxins and PCBs were analysed in both feed and fillet samples according to EU 2017/77 for feed and EU 2017/644 for food.

#### 2.3.4. Sensory Analysis

For sensory analysis, the right fillet was portioned and frozen immediately at −20 °C at the moment of the sampling. A total of 20 frozen portions of 30 g per treatment (PC, 3.5AO and 0.7AO) were analysed for sensory attributes at the CNR-IBE Sensory Laboratory (Bologna, Italy). Samples were stored at room temperature for 90 min, thawed in the microwave and cooked (1100 W; 150 s) before presenting the aluminum-wrapped, warm fish fillets to judges. Sensory profiles of the samples were subjected to descriptive analysis (UNI 10957-2003) quantifying olfactory and gustatory descriptions. A panel test was performed in individual booths (ISO 8589:1988) equipped with netbooks with a specific software (FIZZ, Biosystèmes, France) for sensory data analyses. The CNR-IBE expert panel (10 judges) was trained and selected in accordance with the UNI EN ISO 8586-2:2008. Each of the 10 judges executed two replicates for a total of 20 evaluations. Judges were requested to indicate intensity for each descriptor (olfactory: Overall odor, typical fish odor, freshness; texture: Firmness, fibrosity, adhesiveness, moisture, greasiness; taste/flavor: Sweet, bitter, sour, salty, umami, astringent, overall flavor, typical fish flavor) using a scale from 1 (low intensity) to 9 (high intensity). Lastly, judges evaluated overall sensory quality using the same scale.

### 2.4. Data Analysis

Unless otherwise stated, all statistical tests were performed using R statistical software [42].

#### 2.4.1. Contrasting Theoretically Expected to Actually Observe Nutritional Content of Feeds

The diet formula was compared qualitatively. For the essential fatty acids, i.e., eicosapentaenoic acid (EPA) and docosahexaenoic acid (DHA), individual and total content, the EPA/DHA ratio was compared using a one-way analysis of variance (ANOVA) with diet treatment as an explanatory factor. Individual and total n-3, n-6, and their ratio was treated the same. Significant results were followed up with a Tukey multiple comparison test. The overall fatty acid profiles were compared across diets using an analysis of similarity (ANOSIM) set to 1000 permutations and using the default distance matrix. An analysis of similarity percentage (SIMPER) was used to determine pairwise differences after confirming the differences between experimental treatments using the ANOSIM. In addition, a theoretically expected fatty acid profile estimated from 500 feed analysis for each formula was regressed against the three feed samples taken from each diet in this experiment.

#### 2.4.2. Zootechnical Performance

Survival and all other zootechnical performance parameters (i.e., FCR, SGR, growth, IBW, FBW) were analysed using a one-way analysis of variance (ANOVA) with the diet group as main explanatory variable.

#### 2.4.3. Fatty Acid Profile, Retention, and Lipid Oxidation

Fatty acid profiles were compared using an ANOSIM with five groups (before treatment, NC, PC, 3.5AO, 0.7AO) and a SIMPER to determine pairwise differences. Following the same approach as used for feed, individual and total EPA and DHA, as well as EPA/DHA ratio, were compared amongst the five groups using a one-way ANOVA. Individual and total n-3, n-6, and their ratio, as well as saturated fatty acid (SFA), monounsaturated fatty acid (MUFA), and polysaturated fatty acid (PUFA) content, were analysed the same way. Significant results were followed up with a Tukey multiple comparison test. Differences in lipid oxidation (measured as MDA mg kg^−1^) were analysed using a two-way ANOVA comparing the effect of diet treatment and the difference in measured oxidative stress before and after the feeding trial.

#### 2.4.4. Heavy Metals, Dioxins, and DL-PCBs

The amount of arsenic, mercury, and dioxins or dioxin-like PCBs was compared between the PC diet group and both microalgal diet treatments.

#### 2.4.5. Sensory Data

A one-way analysis of variance (ANOVA) was used with LSD post-hoc analysis using the SAS 7.1 software (Sas Systems, NC).

## 3. Results

### 3.1. Feed Analysis and Comparison between Theoretical and Observed Fatty Acid Composition

All experimental diets contained similar levels of DM (dry matter), protein, and lipids, in order to assure that the nutritional value of each diet was comparable (Table 1). In NC and PC diets, diets formulated with fish oil as the main source of EPA and DHA, the DHA:EPA ratio was 1.54 and 1.37, respectively. Those diets formulated with microalgal oils showed a significantly higher DHA:EPA ratio at above 2 with 2.67 for the 3.5AO and 2.09 for the 0.7AO group (F_3,8_ = 398.18, *p* < 0.001). The 3.5AO diet had a total EPA + DHA content of 2.18%—roughly 2.5-fold greater than that of the 0.7AO or NC diet containing 0.80 and 0.86% EPA + DHA content, respectively (F_3,8_ = 2513.40, *p* < 0.001). The EPA + DHA content of the PC diet was even greater at 3.11%. EPA content ranged from 0.26% in the 0.7AO diet to 1.31% in the PC diet (F_3,8_ = 1255.60, *p* < 0.001), whilst the DHA level was at its lowest with 0.52% in the NC diet but almost similarly high in the PC and 3.5AO diets (F_3,8_ = 3220.10, *p* < 0.001). The total amount of n-3 and n-6 and their ratio also differed statistically among treatments (n-3: F_3,8_ = 319.65, *p* < 0.001; n-6: F_3,8_ = 12.50, *p* = 0.002; n-3:n-6 ratio: F_3,8_ = 13.48, *p* = 0.002).

Additionally, this study recorded a very accurate prediction of the observed fatty acid profile for each experimental diet based on the theoretically expected values (Figure A1, Table A2). The lack of substantial deviation confirmed that feed formulation method retained the promised nutritional profiles for each diet, particularly those including microalgal oils.

### 3.2. Survival and Growth Performance

Survival was high (>99%) amongst all control and experimental diet groups, implying that the feeds were accepted by seabream. In terms of other zootechnical performance measurements (i.e., FCR, SGR, growth, FBW) there was no statistically measurable difference between fish provided with a fish oil or microalgal oil supplemented diet after 110 days of experimental feeding (Table 2). Most importantly, fish from the four experimental treatments duplicated their body weight during the experimental feeding period without any observed side-effects being promoted from either of the diets being tested.

### 3.3. Muscle Fatty Acid Profile, Retention, and Lipid Oxidation

Fillet muscle fatty acid profiles at the beginning of the trial were compared to those measurements taken after 110 days of feeding on different diets and significant differences were found (ANOSIM: Global R: 0.863, *p* = 0.010; Table 3). Specifically, EPA and DHA were highest in the PC with 4.79 g/100 g (F_4,78_ = 249.06, *p* < 0.001, Table 3) and 13.61 g/100 g (F_4,78_ = 71.81, *p* < 0.001), respectively. The ratio of DHA to EPA was 4.15 g/100 g for those fish fed with the 3.5AO—significantly higher than any other diet and from those measurements from before the feeding trial (F_4,78_ = 81.91, *p* < 0.001). The collective DHA and EPA content was highest for fillets of fish fed the PC diet (18.40 g/100 g) closely followed by the 3.5AO diet (12.36 g/100 g). However, both significantly differed from the measurements taken before the feeding trial (10.07 g/100 g) and the NC (7.48 g/100 g) and 0.7AO diets (6.83 g/100 g; F_4,78_ = 91.90, *p* < 0.001). The amount of n-3 and n-6 fatty acids and their ratio also varied amongst groups (n-3: F_4,78_ = 69.01, *p* < 0.001; n-6: F_4,78_ = 545.45, *p* < 0.001; n-3:n-6 ratio: F_4,78_ = 144.83, *p* < 0.001; Table 3). MUFA, PUFA, and SFA differed between diets (MUFA: F_4,78_ = 29.04, *p* < 0.001; PUFA: F_4,78_ = 20.50, *p* < 0.001; SFA: F_4,78_ = 196.47, *p* < 0.001).

In terms of oxidative stress, all groups started with a similar MDA amount before the different dietary treatments commenced (average 1.31 ± 0.24 mg MDA/kg; data not shown). However, there was a significant interaction between dietary treatment and measurements after the feeding trial. The amount of MDA measured after the 110-day experimental feeding (Figure 1) was highest for fish in the positive control, followed by the negative control. Importantly, fish reared on 0.7AO and 3.5AO had significantly lower levels of MDA in their tissue after the feeding trials than those fish fed a diet rich in fish oil (pairwise comparison: PC-0.7AO: *p* = 0.009, PC-3.5AO: *p* < 0.001).

### 3.4. Heavy Metals, Dioxins and DL-PCBs

Dioxin and dioxin-like PCBs were trending to lower quantities in both AO feeds and fillet of fish fed with either AO diet (F_2,8_ = 4.82, *p* = 0.056; Table 4). Arsenic and mercury were detected at comparable quantities taken from the PC, 3.5AO or 0.7AO feed (Table 4).

### 3.5. Sensory

When comparing PC, 3.5AO, and 0.7AO fish fillet samples in terms of gustatory and olfactory preference no significant difference was found (all ANOVA comparisons: F-value ranged between 0.01 to 1.46; *p* > 0.05; Figure 2), implying that the fillets from seabream fed with diets supplemented with either fish oil or microalgal oil were of comparable sensory quality in all attributes.

## 4. Discussion

The use of microalgal oils in the aquafeed industry is considered an alternative to fish oils and nutritionally poorer vegetable oils [29]. However, to reach large-scale usage, microalgal oils must yield products of comparable quantity and nutritional and sensory quality in a variety of economically valuable aquaculture species. The present study expands upon the existing literature [5,35,43,44] on successful replacements of fish oil and vegetable oils with microalgal oil for yet another species, the gilthead seabream (*S. aurata*). Our findings indicate that feeds formulated with 3.5% microalgal oil contain similar levels of n-3 fatty acids. However, a decrease in the level of total EPA + DHA when compared to feeds formulated with 13.7% fish oil needs to be better addressed in future trials to increase comparability amongst diets. Interestingly, one striking but recurring observation when comparing aquafeeds formulated with microalgal oil with those based on fish oil is their nutritional similarities. Omega-3 fatty acid content generally, and DHA content specifically, was similar between aquafeeds containing 13.7% fish oil and those formulated on 3.5% microalgal oil. Dietary DHA is a notoriously scarce fatty acid, but essential for a healthy development, particularly for gilthead seabream [25,38], as this species is unable to biosynthesise this and other essential fatty acids de novo [22,36]. The present study adds to other studies, supporting that marine microalgae are an excellent nutritional alternative to fish oil [29]. In addition, a close match between the expected and actually measured nutritional quality of the feeds was found, providing an important confirmation of the consistency of feed formulation.

The feeding trial in this study confirmed that the inclusion of microalgal oil in diets of gilthead seabream had no effect on any of the zootechnical performance parameters monitored. The results mirror findings from Atlantic salmon (*Salmo salar*; [35]), rainbow trout (*Oncorhynchus mykiss*; [5]), and shrimp (*Litopenaeus vannamei*; [43]). An equal performance is essential to make microalgal oils a viable alternative to fish oil and a better solution than currently provided by vegetable oil, as literature has demonstrated that in gilthead seabream the replacement of 60–100% of fish oil with a variety of vegetable oil lowers growth [18,45,46,47]. Vegetable oil-based diets also demonstrate a vastly different muscle fatty acid profile than the fish oil diet [18,45,47]. Additionally, vegetable oil impacted humoral immunity and induced stress, which may sap metabolic energy that would otherwise be invested into growth [48]. However, in the present study, the MDA content as an indicator of fish stress and welfare, and determinant of product quality and shelf life [49], was lowest for fish supplied with microalgal oil-based diets—much lower than fish oil-fed fish—reinforcing the potential of microalgal oil as a potential substitute for either fish or vegetable oil in aquafeeds.

As already described in sea bream [18,50] and in other species [5,27,44], the fatty acid profile of the feeds was well reflected in the fatty acid profile of gilthead seabream fillets, with high n-3 and DHA contents in fillets from fish fed 3.5% microalgal oil. Additionally, fillets from fish fed 3.5% microalgal oil diets had the second highest EPA and DHA content after the positive control-fed 13.8% fish oil, mimicking the fatty acid composition of the diets, with the highest EPA + DHA levels in the positive control feed. These results are consistent with the fact that gilthead seabream is unable to synthesise large quantities of n-3 long-chain polyunsaturated fatty acids, such as EPA or DHA, from these C-18 fatty acid precursors [22,36]. The data are also in line with other studies reporting comparable levels of EPA and DHA muscle retention: Rainbow trout fed with diets containing 3.5% microalgal oil had identical DHA, EPA, and overall n-3 fatty acid content to those fed with 10% fish oil [5]. Similarly, oil from *Camelina sativa* transgenically engineered to express microalgal genes responsible for synthesising n-3 long-chain polyunsaturated fatty acids achieved high EPA and DHA muscle retention when included in the diets of gilthead seabream [50] and Atlantic salmon [51]. A variety of microalgal diets fed larval gilthead seabream have shown tissue enrichment with DHA even in levels beyond those achieved when fed fish oil [52]. Yet, it is vital to consider the chronicled influences that impact fish muscle fatty acid profiles, such as body stores and feeding history, which contribute to variation between studies [5,21]. Nevertheless, and most importantly, consensus states that replacement of dietary fish oil by microalgal oil results in nutritionally superior fatty acid profiles for the customer than replacement with vegetable oil [53,54].

No differences were found in the levels of heavy metals among treatments. Interestingly, in contrast with specimens supplied feeds formulated using fish oil, specimens reared on feeds using microalgal oils bioaccumulated substantially fewer toxic chemical compounds such as dioxins and PCBs, making those fillets not only nutritionally comparable but also a safer alternative for the customer. This fact is probably related to the absence of fish oil in the diets formulated with microalgal oil, as fish oils used in high energy fish feeds are considered to be the main source of these contaminants in farmed fish [55,56]. In contrast, aquafeeds formulated from microalgae have fewer or completely lack contaminants such as heavy metals, dioxins, or dioxin-like PCBs [57]. Literature has demonstrated that dioxin concentration in feed correlates with the dioxin concentration in the fillet of farmed rainbow trout (*Oncorhynchus mykiss*) [58] and Atlantic salmon (*Salmo salar*) [59,60]. In nature, such toxic contaminants bioaccumulate through the feed web, increasing at every step between primary producers such as autotrophic algae, and end consumers such as large predatory fish [32,33]. Unsurprisingly, fillets from cultured fish fed diets formulated from fish oil or meal contain higher amounts of contaminants than fish fed microalgal-based diets [61]. Here, we reported a lower level of dioxin or dioxin-like PCB contamination in gilthead seabream fed microalgal oil diets. Considering the tolerable weekly intake of a maximum of 2 pg dioxins or dioxin-like PCBs—1 kg recommended by the European Food and Safety Authority—a teenager weighing 50 kg can safely consume three times as much microalgal oil-fed seabream (AO3.5: 680 g; AO0.7: 735 g) than fish oil-fed seabream (PC: 218 g). This makes fish reared on microalgal oil not only comparable in nutritional quality, but safer for consumption. Thus, opportunities exist for downstream value creation through producing healthier and more sustainable aquaculture products to deliver category growth at retail, both in terms of volume and value, contributing to the economic sustainability of aquaculture.

In addition to the increased food safety, from a sustainable perspective, FFDR oil from the diets was decreased to zero, showing an improvement of the marine footprint of these two alternative diets. In order to minimise the usage of fish oil, it is essential that any replacement produces similar end product eating quality. Crucially for the consumer, the present study underscores that gilthead seabream fillets from fish fed with aquafeeds with microalgal oil were liked equally well compared to those fed a diet using fish oil. There was not a single sensory parameter evaluated by the expert panel that significantly differed between dietary treatments. Sensory data obtained from this trial were in agreement with previous sensory data obtained when fish oil was replaced by microalgal oil in salmon diets [62]. This is in contrast with mixed results found for other species: channel catfish (*Ictalurus punctatus*) raised on diets containing dried microalgae had, for example, a slightly fishier flavor [63], whereas Atlantic Salmon fed with diets containing the microalgae *Schizochytrium limacinum* had the same flavour and odour but differed in texture [44]. Since lipid peroxidation can significantly deteriorate seafood sensory quality [49], the low amount of MDA recorded in fillets from fish fed with microalgal oil might have contributed to the comparable quality in this case.

## 5. Conclusions

The present study demonstrated that gilthead seabream fed aquafeeds formulated without fish oil and using 3.5% Veramaris^®^ microalgal oil as a source of EPA and DHA yielded the same growth performance as fish fed aquafeeds formulated with 13.79% of fish oil. The fatty acid profile of the feed was well reflected in the fillet, suggesting that microalgal oil can be successfully used to produce this Mediterranean fish species, maintaining high levels of EPA + DHA in the final product, yet with a lower marine footprint than diets formulated with fish oil. Sensory data indicated that this shift in the oil source can be done without any drawbacks in the sensory quality of the fish fillet. Additionally, the use of Veramaris^®^ microalgal oil decreases the level of contaminants (dioxins and PCBs) in the final product, thus enhancing food safety for the consumer.

## Figures and Tables

**Figure 1 animals-11-00971-f001:**
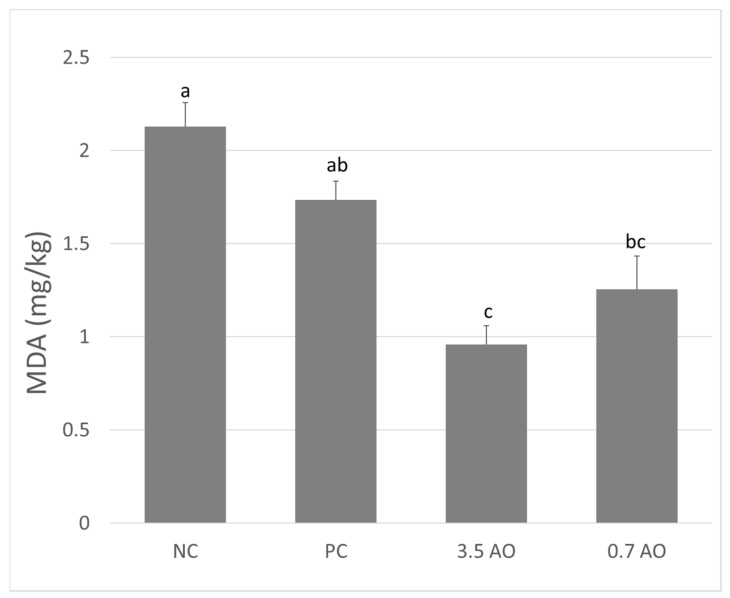
Malondialdehyde (MDA) content in the fillet of gilthead seabream before and after being fed with the different diets. PC: Positive control; NC: Negative control; 3.5AO: 3.5% microalgal oil; 0.7AO: 0.7% microalgal oil. Different letters denote significant differences (*p* < 0.05) between the experimental treatments, while shared letters show no significant difference.

**Figure 2 animals-11-00971-f002:**
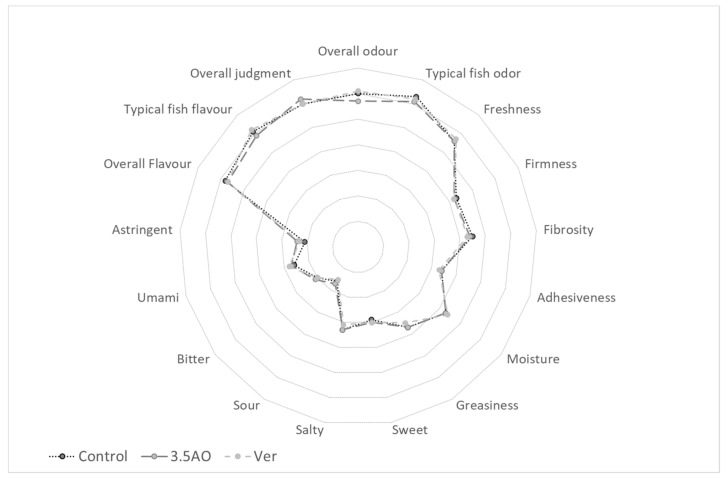
Overlap of sensory description of fillets from gilthead seabream fed with the different diets. C: Positive control; 3.5AO: 3.5% microalgal oil; 0.7AO: 0.7% microalgal oil.

**Table 1 animals-11-00971-t001:** Formulation of the experimental diets. Different letters show significant differences (*p* < 0.05) between the experimental treatments, while shared letters signify no significant difference.

Ingredients (%)	NC	PC		3.5AO		0.7AO
Fish oil	2.10	13.79		0		0
Microalgal oil ^1^	0	0		3.5		0.71
Rapeseed oil	8.47	0		6.09		7.49
Camelina oil	3.22	0		4.20		5.60
Fish meal	19.96					
Soybean meal	8.54					
Guar meal	12.81					
Wheat	10.58					
Corn gluten	22.20					
Wheat gluten	3.77					
Pea	4.27					
DL-methionine ^2^	0.60					
Lysine HCl ^3^	1.02					
Taurine ^4^	0.44					
Monoammonium phosphate ^5^	0.68					
Vitamin C ^6^	0.07					
Vitamin and mineral premix ^7^	1.25					
Protein ^8^	42.96 ± 0.12	43.05 ± 0.15	42.95 ± 0.20		43.03 ± 0.15	
Lipid ^8^	18.0 ± 0.09	18.1 ± 0.07	18.0 ± 0.07		17.9 ± 0.05	
Ash ^8^	4.92 ± 0.07	4.9 ± 0.09	4.93 ± 0.08		4.90 ± 0.09	
Fiber ^8^	2.86 ± 0.08	2.84 ± 0.07	2.82 ± 0.05		2.83 ± 0.09	
Dietary EPA	0.34c	1.31a	0.59b		0.26d	
Dietary DHA	0.52c	1.80a	1.59b		0.54c	
Dietary EPA + DHA	0.86c	3.11a	2.18b		0.80c	
Dietary DHA:EPA	1.54c	1.37d	2.67a		2.09b	
Dietary omega-3	3.83d	4.54c	5.43a		4.79b	
Dietary omega-6	3.84a	2.79b	3.51a		3.85a	
Dietary omega-3:omega-6	1.00c	1.65a	1.55ab		1.25bc	
FFDRoil ^9^	0.60	3.95	0		0	

^1^ VERAMARIS^®^ microalgal oil. ^2^ NOVUS. ^3,6,7^ DSM Nutritional Products. ^4^ Barentz Service. ^5^ Aliphos. ^8^ Analytical data from NIR analyses (*n* = 3). ^9^ FFDRoil = (eFCR × % of fish oil content)/5. Abbreviations: DHA, docosahexaenoic acid; EPA, eicosapentaenoic acid.

**Table 2 animals-11-00971-t002:** Survival and zootechnical performance of gilthead seabream (*Sparus aurata*) fed diets supplemented with either fish oil (negative control (NC), positive control (PC)) or a percentage of microalgal oil (AO) for 110 days.

Diets	NC	PC	3.5AO	0.7AO	Significance
Survival (%)	99.6 ± 0.7	99.6 ± 0.7	99.6 ± 0.7	99.6 ± 0.7	ns
IBW (g)	64.94 ± 1.31	64.19 ± 1.27	64.83 ± 0.91	64.14 ± 1.00	ns
FBW (g)	146.66 ± 2.52	143.95 ± 1.30	145.69 ± 1.26	144.44 ± 3.00	ns
SGR	0.74 ± 0.02	0.73 ± 0.02	0.74 ± 0.01	0.74 ±0.01	ns
FCR	1.44 ± 0.04	1.43 ± 0.05	1.43 ± 0.04	1.44 ± 0.04	ns

Abbreviations: IBW: Initial body weight; FBW: Final body weight; SGR: Specific growth rate (% body weight/day); FCR: Feed conversion ratio (feed intake/weight gain). Data are presented as mean ± SD (standard deviation of 3 replicate tanks (*n* = 3); ns: No significant differences.

**Table 3 animals-11-00971-t003:** Analysed fatty acid (g/100 g) composition of gilthead seabream flesh at the beginning of the trial (TO) and after 110 days of experimental feeding with the different experimental diets.

Fatty Acid		T0	NC	PC	3.5AO	0.7AO
Myristic acid	C14:0	2.48bc	1.44ac	2.50b	1.41a	1.35a
Pentadecanoic acid	C15:0	0.26bc	0.19ac	0.32b	0.19ac	0.18a
Palmitic acid	C16:0	13.02a	11.70a	15.81b	12.63a	11.65a
Palmitoleic acid	C16:1	3.92b	2.41a	4.36b	2.31a	2.36a
Margaric acid	C17:0	0.21ab	0.17a	0.24b	0.15a	0.16a
Heptadecenoic acid	C17:1	0.10a	0.10a	0.18b	0.07a	0.08a
Stearic acid	C18:0	3.17abc	3.00ab	3.23c	2.86ab	3.03ac
Elaidic acid	C18:1	0.10a	0.15a	0.51b	0.09a	0.15a
Oleic acid	C18:1n-9	26.12cd	32.25a	22.79bd	28.43ac	31.80a
Vaccenic acid	C18:1n-7	2.81a	3.08a	2.99a	2.49b	2.92a
Linoleic acid (LA)	C18:2n-6	19.98a	19.01a	12.96b	17.74a	19.23a
α-Linolenic acid (ALA)	C18:3n-3	6.42b	9.78a	4.06b	10.64a	11.36a
Gamma linoleic acid	C18:3n-6	0.44b	0.88a	0.37bc	0.55bc	0.63ab
Octadecatetraenoic acid	C18:4n-3	0.69a	0.80a	1.19b	0.59a	0.58a
Arachidic acid	C20:0	0.23a	0.21ab	0.19b	0.22a	0.23a
Eicosenoic acid	C20:1n-9	1.94d	1.22ac	1.66abd	1.17ac	1.32ab
Eicosadienoic acid	C20:2n-6	0.61a	0.58a	0.45b	0.49a	0.67a
Di-homo-linoleic acid	C20:3n-6	0.34a	0.36a	0.25b	0.29ab	0.32ab
Eicosatrienoic acid	C20:3n-3	0.38a	0.40a	0.23b	0.44a	0.50a
Arachidonic acid	C20:4n-6	0.45a	0.35a	0.71b	0.53a	0.38a
Eicosapentaenoic acid (EPA)	C20:5n-3	3.12b	1.89d	4.79a	2.40c	1.63d
Heneicosanoic acid	C21:0	0.04a	0.04a	0.04a	0.02b	0.04a
Behenic acid	C22:0	0.08b	0.11a	0.10ab	0.11ab	0.13a
Cetoleic acid	C22:1 n-11	1.45b	0.84a	1.79b	0.73a	0.72a
Erucic acid	C22:1n-9	0.64b	0.43a	0.45a	0.37a	0.44a
Docosadienoic acid	C22:2n-6	0.78b	0.67ac	0.86bc	0.64ac	0.65ac
Docosapentaenoic acid (DPA)	C22:5n-3	1.95b	1.13a	1.96b	1.33a	1.05a
Docosahexaenoic acid (DHA)	C22:6n-3	6.95c	5.58cd	13.61a	9.97b	5.20d
Tetracosanoic acid	C24:1n-9	0.57abc	0.48ac	0.65b	0.42a	0.47ac
Pentacosaenoic acid	C25:0	0.04b	0.07a	0.07a	0.07a	0.07a
EPA + DHA		10.07c	7.48d	18.40a	12.36b	6.83d
DHA: EPA		2.23d	2.95bc	2.84c	4.15a	3.19b
Total omega-3		20.22b	20.28b	26.54a	26.03a	21.01b
Total omega-6		22.59a	21.85b	15.59d	20.24c	21.88b
Omega-3:omega-6		0.90c	0.93c	1.71a	1.29b	0.96c
SFA		19.48b	16.89d	22.46a	17.65c	16.80d
MUFA		37.67b	40.95a	35.37c	36.06bc	40.27a
PUFA		42.81b	42.12b	42.13b	46.26a	42.89b

Abbreviations: ALA, α-linoleic acid; DHA, docosahexaenoic acid; DPA, docosapentaenoic acid; EPA, eicosapentaenoic acid; LA, linoleic acid; SFA, saturated fatty acids; MUFA, monounsaturated fatty acids; PUFA, polyunsaturated fatty acids. Note: Different letters show significant differences (*p* < 0.05) between the experimental treatments, while shared letters signify no significant difference.

**Table 4 animals-11-00971-t004:** Amount of heavy metals, dioxins, and dioxin-like polychlorinated biphenyls (DL-PCBs) found in feed and gilthead seabream fillets.

	PC		3.5AO		0.7AO	
	Feed	Fillets	Feed	Fillets	Feed	Fillets
DL-PCB and dioxins (pg/g wet weight)	0.464	0.455 ± 0.230	0.298	0.145 ± 0.073	0.307	0.131 ± 0.066
Arsenic (mg/Kg)	0.77	0.81 ± 0.24	0.6	0.75 ± 0.23	0.67	0.68 ± 0.20
Mercury (mg/Kg)	0.032	0.039 ± 0.012	0.03	0.060 ± 0.018	0.032	0.051 ± 0.015

For fillets’ standard deviation (±) calculated from 3 replicate tanks (*n* = 3).

## Data Availability

Data will be uploaded to an online repository upon acceptance of the manuscript.

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
