# Peer review of "Microalgae Oil as an Effective Alternative Source of EPA and DHA for Gilthead Seabream (Sparus aurata) Aquaculture"

_animals, 2021, doi:10.3390/ani11040971_

Round 1

Reviewer 1 Report

Dear Authors,

thank you for taken into account of my previous comments and suggestions on your manuscript, I found your revisions almost totally satisfactory, and the manuscript is now more accurate, clear and enhanced.

However there are still some minor revisions to address before the publication in my opinion, you can find my new comments in the attached report.

Best regards

The reviewer

Reviewer 2 Report

 Accept in present form

Author Response

Thanks for your comments and suggestions in the first round that helped to improve the quality of the manuscript.

Reviewer 3 Report

This study explores the use of microalgae oil as an alternative supplement to fish feed compared to traditional fish oil-based feeds. The study examines a range of parameters, including (and focusing on) the uptake and retention of key fatty acids EPA and DHA in gilthead seabream, the uptake of toxins, and the impact on flesh quality. 

Overall the manuscript is well-written and well-presented. Analyses and figures/tables appear appropriate, and the design of the study is also appropriate to test the hypotheses. I think the methods are complete, and it seems fish cared for appropriately. Whilst it only represents an iterative contribution to the field, I think it does make some important strides and would be a valuable publication. I have few comments on the manuscript, presented below:

MAJOR

  1. Table 1: Percentage of rapeseed and camelina oil varies between diets. I imagine this is to help ensure diets are isoenergetic, isonitrogenous and isolipidic (Lines 127-129) but are these otherwise inert? Will these differences influence the nutritional quality? I suppose this may be partially controlled for (negative control), but how confident are the authors about the impact of their contribution compared to AO?
  2. Lines 407-430: This paragraph deals with food safety and the reduced uptake of PCBs and dioxins into fillets. However, little mention is made of arsenic and mercury. It appears a considerably greater proportion of feed-based arsenic is retained within fillets than in PC fish. Whilst there is relatively large variance, similarly there is an increase in concentration of mercury in AO versus PC fish. This does need to be mentioned in the Discussion. For the results, it is stated that "Dioxin and dioxin-like PCBs were strongly trending to drastically lower quantities in both AO feeds and fillet of fish fed with either AO diet". However, there is substantial variance, and the omnibus analysis suggests only weak evidence against the null hypothesis (F=4.8, p=0.056). So I think this needs to be interpreted positively but a little bit more cautiously.
  3. Lines 431-446: This paragraph deals with food sustainability and quality. Just out of interest, is there any idea what causes the variation in findings regarding flesh quality? Is it the fish, or do different microalgae impart different qualities?

MINOR

  1. Line 42: What challenges? You haven't indicated these yet
  2. Line 76-78: [sentence beginning "Because..."] This sentence is unfinished as written. Should it be a comma before "Because"?
  3. Line 95: This is the first instance of the abbreviation AO in the main text (i.e. outside of the Abstract) but it is undefined; make sure you indicate it refers to microalgal oil somewhere around here.
  4. Line 119-123: What was the basis for these particular concentrations of fish oil and AO? 13.79% fish oil is justified as mimicking a commercial diet but it would be useful to know how other concentrations were chosen.
  5. Line 180: fillets were frozen. Was this flash-freezing? At what temperature were they maintained? These were used for determining fatty acid composition but were a subset of these used for sensory analysis? If not, how were the portions for sensory analysis collected and maintained?
  6. Line 451-454: I don't think it is correct or appropriate to suggest high levels of EPA in the fish fed the AO diet, since EPA levels were lower than the positive control and, importantly, had reduced from the pre-trial concentrations.
  7. General: Does method of formulation impact bioavailability of supplements? 

Author Response

This manuscript is a resubmission of an earlier submission. The following is a list of the peer review reports and author responses from that submission.

Round 1

Reviewer 1 Report

This article is interesting and the scientific methodology is robust, the experimental plan is well conceived, therefore this article is worth of publication.

However the authors must include some clarifications both substantial then formal.

 One of the two low level of algal oil inclusion is terribly low: 0.7%. This level of inclusion is really difficult to be justified in a scientific experimentation on fish nutrition. I don’t see any possible reason for that. The authors must to say the reason for that. I suppose that it can be expensive ingredient, or a previous experiment, but they have to explain this point.

The authors must introduce a comment about economic sustainability of this algal oil inclusion in the fish feeds.

Table 1.

Authors should include feeds gross energy, even inferred by calculation if not measured in laboratory.

Feeds proximate analyses must be separately indicated for each feed, possibly including ash content. Data must be expressed as average and standard   deviation.

Did you make any fillet analyses in an intermediate moment of the experimentation, or only in the beginning and the end of the experiment? This could be useful to investigate the trend of fillet composition.

At lines 363 – 366 Authors wrote:

“In contrast with specimens supplied feeds using fish oil, specimens reared on feeds using microalgal oils bioaccumulated substantially fewer toxic chemical compounds, making those fillets not only a nutritionally comparable but also safer alternative for the customer.”

They should give an explanation of this phenomenon, even hypothetical.

There are English mistakes and some repetitions. An English revision is requested.

Reviewer 2 Report

Dear Authors,

after an accurate analysis of your manuscript, my opinion is to reject it in the present form. 

This is principally due to the bad writing and drafting quality of all text, (especially in Introduction and Discussion sections that are really confused and not very fluent for the readers) that before an eventual new submission, must be subjected to careful editing by a language expert.

Moreover, unlike what you stated in the manuscript, almost all the research questions of your study are already satisfied by other published manuscript. Through the most common scientific search engines, I found manuscript that have already covered the same topic, for example:

E. Ganuza, T. Benítez-Santana, E. Atalah, O. Vega-Orellana, R. Ganga, M.S. Izquierdo,
Crypthecodinium cohnii and Schizochytrium sp. as potential substitutes to fisheries-derived oils from seabream (Sparus aurata) microdiets,
Aquaculture,
Volume 277, Issues 1–2,
2008,
Pages 109-116,
ISSN 0044-8486,
https://doi.org/10.1016/j.aquaculture.2008.02.005. 

Marta Carvalho, Daniel Montero, Grethe Rosenlund, Ramón Fontanillas, Rafael Ginés, Marisol Izquierdo,
Effective complete replacement of fish oil by combining poultry and microalgae oils in practical diets for gilthead sea bream (Sparus aurata) fingerlings,
Aquaculture,
Volume 529,
2020,
735696,
ISSN 0044-8486,
https://doi.org/10.1016/j.aquaculture.2020.735696.

E. Atalah, C.M. Hernández Cruz, M.S. Izquierdo, G. Rosenlund, M.J. Caballero, A. Valencia, L. Robaina,
Two microalgae Crypthecodinium cohnii and Phaeodactylum tricornutum as alternative source of essential fatty acids in starter feeds for seabream (Sparus aurata),
Aquaculture,
Volume 270, Issues 1–4,
2007,
Pages 178-185,
ISSN 0044-8486,
https://doi.org/10.1016/j.aquaculture.2007.04.009.

However, trying to help you in any future resubmission, I report futher weakness of your manuscript that I found, that should be revised.

Simple Summary and Abstract sections are really long and exceed the limits imposed by the journal reported in the Istructions for Authors (maximum 200 words). Moreover, these sections are really vague and reported few key data of your study, that are the primary interest of the readers.

Between Keywords is better to avoid the repetition of words already reported in the Title, in order to give more resonance to your manuscript during the searching phases on web. Four out of six of yours are repeated.

Line 95 (..and other similar points in the text): why you reported as examples (e.g.) references which you support your sentences? It aren't examples, are references.

You should be more accurate in Material and Methods section, regarding the formulation and the relative percentage of each component (why are the formulations totally different, and not only for the tested components? On what empirical bases did you rely for this formulations?), and moreover was not mentioned the shape of the food and the relative size.

On the line 135 you write that the refused or excess food, was collected and quantified, how did you manage to do it precisely?

Why did you choose to sacrificed the specimens by a blow on the head after anesthetization, and not with a normal euthanasia with a letal dose of anesthetic?

2.4 Data analysis, is very approximate, with the sentence "Unless otherwise stated all statistical tests were performed using R statistical software". All technical procedures should be well specified.

In the main text you refer to a "15 weeks of experimental period", but in Table 2 you reported a time of "110 days", that are not exactly the same. Please use only accurate one way to refer to this key data of the study.

The quality of the Figures and Graphics must be improved. In Figure 2 you reported a strange result; why before the different feeding treatment, the MDA content was different among the experimental groups? What is it due to?

Moreover, why you have not chosen to test also other similar biomarkers as: SOD and CAT value or enzymatic activity or cortisol level?

4. Discussion has been lost, I suppose that Discussion section should starts around line 350.

The sentence that you reported between lines 363-366 should be supported by more specific analysis, dedicated to this aspect, or avoided in the present form.

Line 388: "..the replacement of between 60±00% of fish oil.." what do you mean?

The results of your study are in my opinion non well discussed and reported in the Conclusion section. Total content of DHA + EPA results higher in PC group, this data should be better related with the relative content of fish oil in the formulation compared to microalgal oils in the others. This relative comparison was missing in your manuscript or not well highlighted for the reader. This is an important data in my opinion, to better understand or proposed in the Conlcusion section any fish oil replacement precentage with microalgal oils.

Please double check that each key statements of the manuscript, is adequately supported by references.

Moreover, double check that in the references all the scientific names of the living organisms are in italics.

Best regards 

The reviewer

Reviewer 3 Report

The authors in their manuscript “Microalgae oil as an effective alternative source of EPA and DHA for gilthead seabream (Sparus aurata) aquaculture” present a very well-organized work. Strong evidence through experimental data was provided to support the results. I will accept after minor revision to manuscript.

Please find below some suggestions for improving the quality of the manuscript:

  • EPA + DHA and EPA+DHA; DHA:EPA and DHA: Correct in Introduction.
  • Table 1, column 1: Line: EPA, DHA….. Clarify if it refers to microalgae oil.
  • The conclusion is not fully supported by the results. Besides minimizing contamination, what it the cost to aquaculture of using microalgae oil instead of fish oil.
  • Include as appendix the polyunsaturated fats chromatograms of muscle.
